# A multiplexed, confinable CRISPR/Cas9 gene drive can propagate in caged *Aedes aegypti* populations

Michelle A. E. Anderson [1,4,8], Estela Gonzalez [1,5,8], Matthew P. Edgington [1,4], Joshua X. D. Ang[1,4], Deepak-Kumar Purusothaman [1,6], Lewis Shackleford[1,4], Katherine Nevard [1], Sebald A. N. Verkuijl [1,2], Timothy Harvey-Samuel[1], Philip T. Leftwich [1,7], Kevin Esvelt[3] & Luke Alphey [1,4] ✉

*Aedes aegypti* is the main vector of several major pathogens including dengue, Zika and chikungunya viruses. Classical mosquito control strategies utilizing insecticides are threatened by rising resistance. This has stimulated interest in new genetic systems such as gene drivesHere, we test the regulatory sequences from the *Ae. aegypti benign gonial cell neoplasm* (*bgcn*) homolog to express Cas9 and a separate multiplexing sgRNA-expressing cassette inserted into the *Ae. aegypti kynurenine 3-monooxygenase* (*kmo*) gene. When combined, these two elements provide highly effective germline cutting at the *kmo* locus and act as a gene drive. Our target genetic element drives through a cage trial population such that carrier frequency of the element increases from 50% to up to 89% of the population despite significant fitness costs to *kmo* insertions. Deep sequencing suggests that the multiplexing design could mitigate resistance allele formation in our gene drive system.

The highly invasive nature of *Ae. aegypti* and its rapid adaptation to human-commensal habitats such as densely-populated cities/towns have played a significant role in the global spread of vector-borne diseases[1,2]. With up to 40% of the world's population at risk of infection, an estimated 390 million infections per year for dengue alone[3], and a predicted increase in the future due to climate change and urbanization[4], control of the *Ae. aegypti* vector is fundamental to reducing this burden[2]. In the past, conventional control methods have successfully suppressed mosquito populations and the associated burden of disease[2]. However, the inherent limitations and reductions in efficacy brought about through insecticide resistance and off-target impacts have highlighted the need for research into orthogonal, effective, and environmentally friendly alternatives - including gene drives[5].

Gene drives are a means of biasing inheritance to spread a trait of interest through a target vector population[6,7]. The development of the readily programmable nuclease, CRISPR/Cas9 greatly facilitated the development of homing-based drives, with a focus on their potential for mosquito control[6,8]. These gene drives consist of a Cas9 endonuclease and at least one programmable single guide RNA (sgRNA), which directs the Cas9 to the gDNA at the target site. The Cas9 then creates a double-stranded break which must be repaired by the cell's DNA repair machinery. By targeting a site at which components of the gene drive have been inserted into the genome, ideally homology-directed repair (HDR) will result in conversion of the cut allele into a gene drive carrying allele through a process referred to as homing. Alternatively, the cut may be repaired by non-homologous end joining (NHEJ). There have been demonstrations of the remarkable efficiency

[1]Arthropod Genetics, The Pirbright Institute, Ash Road, Pirbright GU24 0HN, UK. [2]Department of Biology, University of Oxford, 11a Mansfield Road, Oxford OX1 3SZ, UK. [3]Media Laboratory, Massachusetts Institute of Technology, Cambridge, MA 02139, USA. [4]Present address: The Department of Biology, University of York, Wentworth Way, York YO10 5DD, UK. [5]Present address: Animal and Plant Health Agency, Woodham Lane, Addlestone, Surrey KT15 3NB, UK. [6]Present address: MRC-University of Glasgow Centre for Virus Research, Henry Wellcome Building, 464 Bearsden Road, Glasgow G61 1QH, UK. [7]Present address: School of Biological Sciences, University of East Anglia, Norwich Research Park, Norwich, Norfolk NR4 7TJ, UK. [8]These authors contributed equally: Michelle A. E. Anderson, Estela Gonzalez. ✉e-mail: luke.alphey@york.ac.uk

of Cas9-based homing drives at biasing inheritance in a few organisms, namely the yeast *Saccharomyces cerevisiae* and *Anopheles stephensi*, *An. gambiae* mosquitoes[9–11]. However, this high efficiency has not been replicated in other species such as *Drosophila melanogaster, Ae. aegypti* and mammals such as mice[12–14].

Like any pest management intervention, gene drives will select for resistance in the target organism. Sequence variations in the target loci caused either by pre-existing heterogeneity or mutations induced through cut-site repair without homing, may lead to the selection of 'resistance alleles'[15–17]. These resistance alleles can rapidly accumulate in the population if they also maintain the function of the target gene, so-called 'r1' alleles (as opposed to 'r2' which are resistant to cleavage but non-functional). In some of the first gene drives tested in cage trials, resistance led to the rapid inhibition of homing and drive[15,18], a problem that remains to be overcome. Including multiple sgRNAs targeting numerous sequences at the target loci or 'multiplexing' is one potential way to mitigate against this[8]; pre-existing sequence variations (whether r1 or r2) or failed homing attempts must have inhibited all target sequences to fully prevent further drive[19–24]. Despite the theoretical viability of the multiplexing approach, in early work in *D. melanogaster* multiplexed systems were not successful[19]. However, later refinements in the specifics of the designs such as targeting a haplolethal gene were successful in caged trials[21]. Such a system is furthermore less likely to result in functional resistant (r1) alleles, given the multiple disruptions to the target gene, and will function most effectively if non-functional mutations result in some fitness cost. Close linkage of the sites may be necessary for HDR efficiency, minimizing the sequence length which must be resected however, there is a possibility that NHEJ-based repair at one site may affect the target sequence of closely linked sites, for example if a large deletion is caused[25]. Here we investigate the feasibility of a multiplex design in *Ae. aegypti*.

One of the most attractive features of CRISPR/Cas9 gene drives is their potential to spread from very low initial release frequencies[6], but this efficiency is also a cause for concern. The dangers of accidental release or issues around control in the field have promoted interest in less invasive, threshold-dependent gene drive systems that are more geographically confinable ("localized")[26–31]. Split-drive systems, where one essential component of the drive does not itself benefit from biased inheritance, allow for safe and straightforward optimization and comparison of the different components of the drive, and provide many of the desirable effects of CRISPR/Cas9 homing gene drives with increased control[5–8]. While non-localized gene drives have been tested in a handful of dipteran species[10,12], population-level assessment of confinable 'split-drive' designs has previously only been empirically demonstrated over multiple generations in *D. melanogaster*[32].

A split-drive system requires separating the drive into two parts, fortunately, CRISPR/Cas9-based drives provide a natural split: the Cas9 protein and an sgRNA that defines the target sequence. Part of the drive – the component that will home and correspondingly benefit from biased inheritance – is inserted into the target region where the Cas9 will cut, guided by the sgRNAs designed explicitly for that region. We selected *kmo*, an attractive target gene, for initial gene drive studies and designed a multiplex homing cassette expressing four sgRNAs targeting the *kmo* gene (hereafter referred to as *kmo*^sgRNAs). *kmo* is required for the synthesis of ommochrome pigments in mosquitoes; homozygotes for non-functional mutant alleles display a white eyed phenotype[33,34]. The recessive eye phenotype allows easy tracking of insertional mutants (also marked by a fluorescent protein), and other non-functional mutations resulting from NHEJ[33]. Mosaicism observed in the eye can be a useful indication of somatic expression or deposition of Cas9/sgRNAs into the embryo[33,34].

Indels generated in somatic tissues could result in a phenotype similar to homozygotes if cut rates are high in the relevant tissues. In drives that target haplosufficient female sterility genes this could result

in females heterozygous for the drive elements becoming sterile themselves[10,35]; somatic expression of active nuclease in heterozygotes is therefore undesirable. The second element of the split-drive assessed here utilizes the regulatory elements from *Ae. aegypti bgcn* to express Cas9. *bgcn* has been identified and characterized as a regulator of cystoblast formation in *D. melanogaster*; transcripts are restricted to a few cells, including germline stem cells[35]. This restricted expression pattern is favorable for confining Cas9 expression to the germline and minimizing somatic expression/cutting[36].

Each element on its own will be transmitted between generations under standard Mendelian principles and rates of inheritance. However, when these two components come together in a single organism, the desired outcome is cleavage of *kmo* in the germline, allowing the *kmo*^sgRNAs element to be utilized as a template for HDR and so bias inheritance in its favor ("drive"). In simple crosses between trans-heterozygotes (*kmo*^sgRNAs; *bgcn*-Cas9) and WT, we observed an inheritance rate of the *kmo*^sgRNAs element of greater than 75%. In our small cage trials, we observed highly effective germline cutting rates. The split-drive was able to bias inheritance such that, after several generations, up to 89% of a population carried the element. These results demonstrate the ability of this proof-of-principle split, multiplexed CRISPR/Cas9 homing drive to increase in frequency within *Ae. aegypti* populations over multiple generations and validate previous modelling work predicting the general dynamics of this type of system.

## Results
### Design and generation of split-drive elements
It has been proposed that multiplexed designs may mitigate the formation and accumulation of resistance alleles due to the use of multiple target sites. To investigate this hypothesis we designed an array of four different RNA pol III promoters, each expressing a different sgRNA targeting four, closely linked, sequences in *kmo* (Fig. 1a). *Ae. aegypti* endogenous pol III promoters were selected based on expression in *Ae. aegypti* Aag2 cells and *Ae. aegypti* transgenics[37,38]. These four promoters were all highly active in Aag2 cells, in rank order of U6.702, 7SK, U6.774, U6.763[37]. The three U6 promoters were assessed by Li et al. for their ability to generate germline mutations in the *white* gene. In those experiments U6.763 gave the most germline mutants followed by U6.774 then U6.702[38]. Three previously verified[39] and one new sgRNA target were selected within a region of approximately 135 bp, each are expressed with one of four unique backbone variants[40] to minimize repetitive sequences within the construct. sgRNA519 and 468 were the most effective, as determined by HRMA analysis of the *kmo* gene in injected embryos, with 447 being the least effective by comparison. In deciding on the combination of promoter and sgRNA, we attempted to average out these differences, combining the most effective promoter with the least effective sgRNA, etc – with the caveat that one promoter (7SK) and one sgRNA (499) were untested in vivo. The 7SK promoter was highly active in Aag2 cells[37] and so we paired it with the most active previously assessed sgRNA 468. The most active in vivo promoter U6.763 was paired with the lowest ranking sgRNA. The next most active promoter, U6774 was paired with the second most active sgRNA 519. And lastly, U6.702 which was the least active promoter in vivo but the most active in Aag2 cells, was paired with the untested sgRNA 499. This plasmid (AGG1095) uses a 1.2 kb 5' homology arm and a 1.9 kb 3' homology arm to integrate the multiplexed sgRNA array and an AmCyan fluorescent marker into the genome. The homology arms exclude this 135 bp of *kmo* exon five, which contains all four sgRNA target sites (Fig. 1a, Table S1), such that these are absent from the drive allele. It should be noted that this 135 bp region includes sequence beyond the cut sites of even the outermost sgRNAs (Fig. 1a, top line), such that even the end sgRNAs do not have precise homology at either end. This avoids the outermost sgRNAs having a privileged position relative to the internal ones (without the requirement for resection). It has previously been

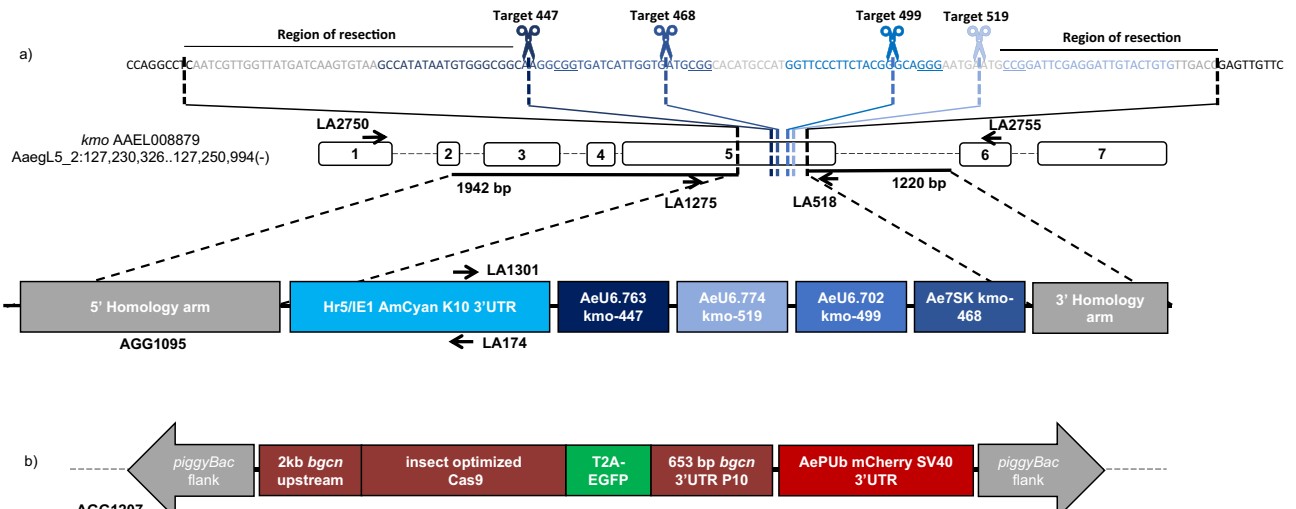

**Fig. 1 | Split-drive designs. a** Four sgRNA targets were selected within exon 5 of the *kmo* (AAEL008879) gene (dashed blue lines) of *Ae. aegypti*, disruption of which results in loss of pigmentation in the eye. The 135 bp region containing all four targets which is excluded from the plasmid (nucleotides between black dashed lines indicated in grey), is depicted above with the cut sites of each sgRNA indicated by scissors and blue text, PAM (protospacer-adjacent motif) sites are underlined. Note that cuts at any of the sgRNA target sites would require resection to reach perfect homology. Upstream and downstream regions were sequence confirmed in the Liverpool strain (WT) and the region of homology included in the plasmid is indicated by black bars. AGG1095 plasmid consists of an Hr5/IE1 (AcMNPV, ie1

promoter fused with homologous region 5 enhancer) expressing AmCyan (*Anemonia majano* cyan fluorescent protein) fluorescent marker, and four *endogenous Ae. aegypti* RNA pol III promoters each expressing the specified sgRNA. **b** The *bgcn*-Cas9 expression plasmid uses -2 kb upstream sequence of the *bgcn* gene to express an insect codon optimized Cas9 followed by T2A-EGFP (2A peptide from foot-and-mouth-disease virus, enhanced green fluorescent protein) and the *bgcn* 3′UTR and an additional P10 3′UTR (*Autographa californica* nucleopolyhedrovirus P10 3′UTR). A PUb-mCherry marker also contained within the *piggy*Bac transposable element flanks allows for identification of transgenic mosquitoes. Individual elements are not to scale.

demonstrated in *D. melanogaster* that this multiplexing design led to a dramatic reduction in drive efficiency[41]. Here we wished to determine the feasibility of this strategy.

Embryonic microinjection with in vitro transcribed sgRNAs, plasmid AGG1095, and Cas9 protein generated several transgenic lines positive for the fluorescence marker (Table S1). Integration into the *kmo* gene was confirmed by PCR (Fig. S4). All further investigations were carried out using a line derived from a single PCR confirmed $G_1$ male ($kmo^{sgRNAs}$).

Embryonic microinjections into Liverpool strain ('WT') with the *bgcn*-Cas9 (AGG1207) construct and *piggyBac* transposase (Fig. 1b) yielded at least five insertion events (Table S1). These five transgenic lines were assessed for the absence of sex-linkage, multiple insertions, and homozygous viability (Table S2) and one was selected to determine its ability to bias the inheritance of the $kmo^{sgRNAs}$ element in a standardized series of crosses.

### Determination of inheritance bias by *bgcn*-Cas9

For the selected *bgcn*-Cas9 insertion line (D), we first crossed *bgcn*-Cas9 females to $kmo^{sgRNAs}$ males and termed this the $F_0$ cross (Table S3). Trans-heterozygous ($kmo^{sgRNAs}$; *bgcn*-Cas9) $F_1$ progeny were then crossed to Liverpool WT of the opposite sex, and their progeny scored for inheritance of the $kmo^{sgRNAs}$ element (Fig. 2a, Table S4). These progeny ($F_2$) were collected in pools, separately for each lineage of crosses. We observed super-Mendelian inheritance of the $kmo^{sgRNAs}$ (G-test: $G_1 = 90.875$, $p < 0.001$, Fig. 2a, Tables S4 and S5). For this line, which showed evidence of inheritance bias for both sexes (68% in males G-test: $G_1 = 15.221$, $p < 0.001$; 77% in females $G_1 = 98.201$, $p < 0.001$, Table S6), we next set out to more accurately quantify the rate of inheritance bias, the overall germline cutting rate and relative contribution of the individual sgRNAs.

Firstly, we repeated the cross between trans-heterozygous $kmo^{sgRNAs}$; *bgcn*-Cas9 females and WT males, this time collecting the $F_2$ progeny separately from each female. We screened for the rate of inheritance of the $kmo^{sgRNAs}$ element; 81.2% ([approx. 95%

CI] = [78.5–83.6%], as well as eye phenotype as a measure of cutting rate (83% [80.5–85.3%]) (Fig. 2b, Table S7). In this case we found the majority of cuts resulting in HDR repair and only a few percentage points being repaired by NHEJ.

### Determination of inheritance bias and cutting rates using the split-drive system

To assess the overall cutting and efficiency of the drive to bias inheritance, trans-heterozygous $kmo^{sgRNAs}$; *bgcn*-Cas9 ($F_0$ *bgcn*-Cas9 females crossed to $kmo^{sgRNAs}$ males) themselves having a mosaic phenotype were crossed to a gene-edited *kmo* knock-out line ($kmo^{-/-}$) as single pair crosses and the progeny of each individual cross was scored separately (Fig. 2d). The offspring of this cross were screened for AmCyan fluorescence, indicating the inheritance of the $kmo^{sgRNAs}$ allele, and for eye phenotype. The drive was inherited by 77.2% ([approx. 95% CI] = [66.8–85.1%]) of the progeny of the male trans-heterozygotes and 75.7% [65.5–83.6%] of the progeny of the female trans-heterozygotes (Fig. 2d, Tables S8, S9 Model 1), substantially higher than predicted odds from Mendelian inheritance rates of 50% (Binomial GLMM: Log-Odds = 1.18 [0.82–1.53], $p < 0.001$, Table S9 Model 2), and with no significant effect of parental origin for the Cas9/drive allele (Binomial GLMM: Odds ratio = 0.92 [0.45–1.88]), $p = 0.816$, Table 9 Model 1, Fig. 2d). These estimates, which incorporated batch effects, were slightly elevated from the pooled data, especially for males (males 71.4% [69–73.8%], females 72.9% [70.2–75.4%]) (Fig. 2a, Table S9 Model 4), indicative of a significant level of individual variation in efficiency. As a comparison, the inheritance of the Cas9 allele (which should conform to standard Mendelian inheritance), was 48.8% for males, and 50.1% for females (Binomial GLMM: $\beta = -0.05$ [−0.2–0.1], $p = 0.474$; $\beta = 0.05$ [−0.16–0.26], $p = 0.588$), and indicates no major effect on viability of the *bgcn*-Cas9 allele under these conditions.

In this experimental design the only functional *kmo* allele is in the chromosome homologous to the $kmo^{sgRNAs}$ allele in the trans-heterozygous parent. Progeny of this cross which lack the cyan fluorescence marker indicative of the $kmo^{sgRNAs}$ element must have inherited

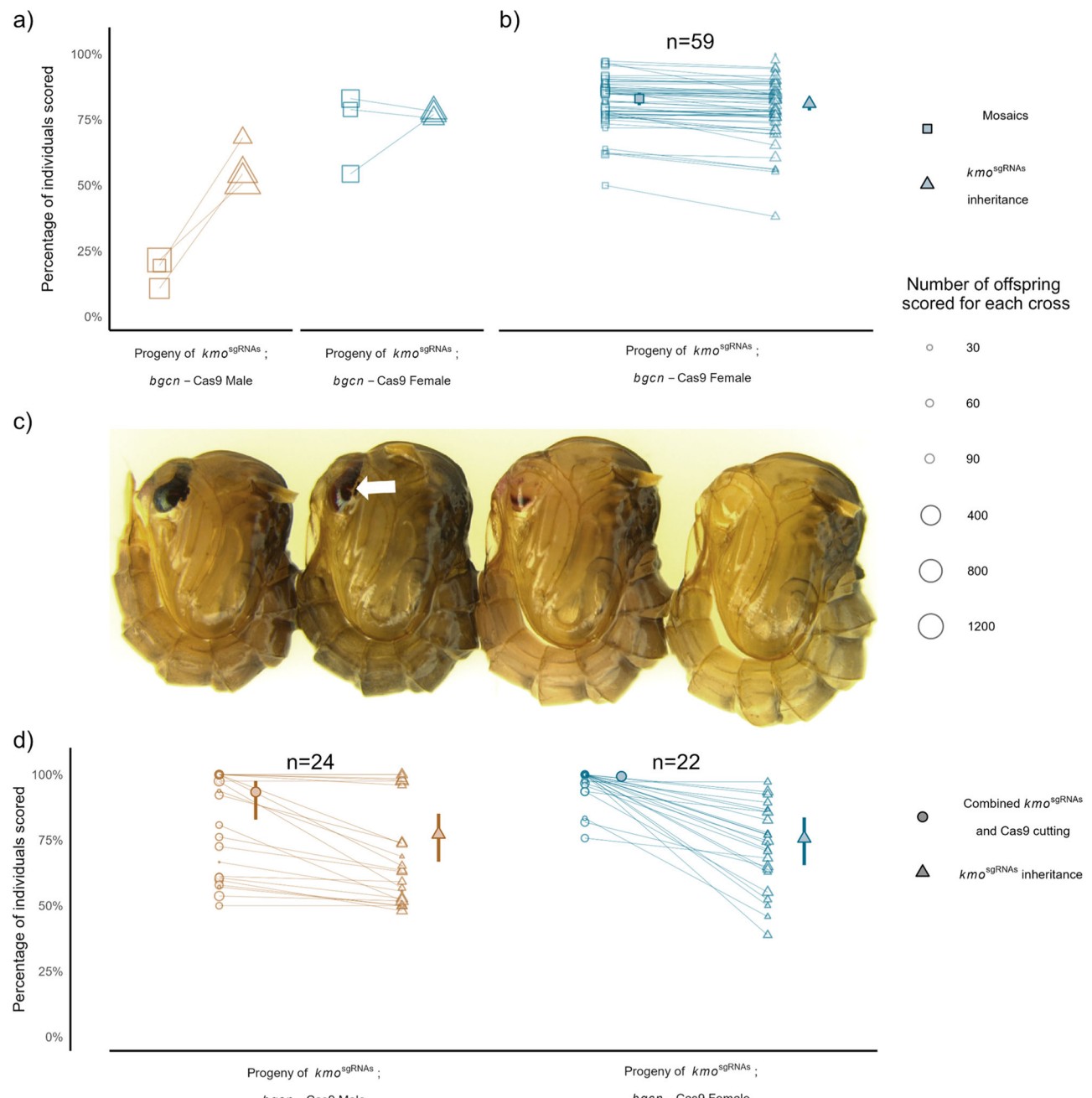

**Fig. 2 | *bgcn*-Cas9 biases the inheritance of *kmo*sgRNAs. a** Female *bgcn*-Cas9 were crossed to *kmo*sgRNAs males (F$_0$) for an initial pooled assessment. Trans-heterozygous F$_1$ males and females were outcrossed to WT in three replicate crosses and the F$_2$ progeny scored for inheritance of the *kmo*sgRNAs transgene (triangles) and eye phenotype (squares). **b** A replicate cross was completed and the proportion of the F$_2$ progeny inheriting the *kmo*sgRNAs transgene (triangles) and eye phenotype (squares) were scored from individual F$_1$ females. **c** Images of pupae displaying the different eye phenotypes from left to right: wild type (dark), weak mosaic (white arrow indicating mosaicism), strong mosaic, white eye. **d** Combined *kmo*sgRNAs inheritance and Cas9 cutting are represented by circle points and are measured as the percentage of offspring with white eyes from crosses between

males (*n* = 24) or females (*n* = 22) trans-heterozygous for *kmo*sgRNAs; *bgcn*-Cas9 and *kmo*$^{-/-}$ mosquitoes. *kmo*sgRNAs inheritance alone is measured as the percentage of offspring with AmCyan fluorescence (triangles). Individual faded points represent offspring from one drive parent, and the size of the point is proportional to the number of offspring from the parent. Horizontal lines connect pools of offspring to show the relationship between cutting and inheritance rates in each replicate. Large symbols and error bars (vertical lines) represent estimated mean and 95% confidence intervals calculated by a generalized linear mixed model, with a binomial ('logit' link) error distribution. Data points may be overlapped and some cannot be discerned. Source data are provided as a Source Data file.

this other, nominally wild type *kmo* allele. In such individuals, completely white eyes indicate that this allele has been mutated through cutting and error-prone repair. This repair likely occurs either in the germline of the trans-heterozygous parent or very early in the developing zygote, or in principle, one or more later events that still affect all relevant cells providing pigmented ommatidia. Mosaic eyes indicate

non-functional mutations generated later in the developing zygote, such that some cells forming ommatidia have *kmo* function, but others do not (Fig. 2c, Table S8). We observed white eyes in F$_2$ progeny at a rate of 93.4% [82.8–97.6] from F$_1$ males and 99.3% [97.5–99.8%] from F$_1$ females indicating that while germline/early zygotic cutting efficiencies are very high in both sexes, offspring of trans-heterozygous

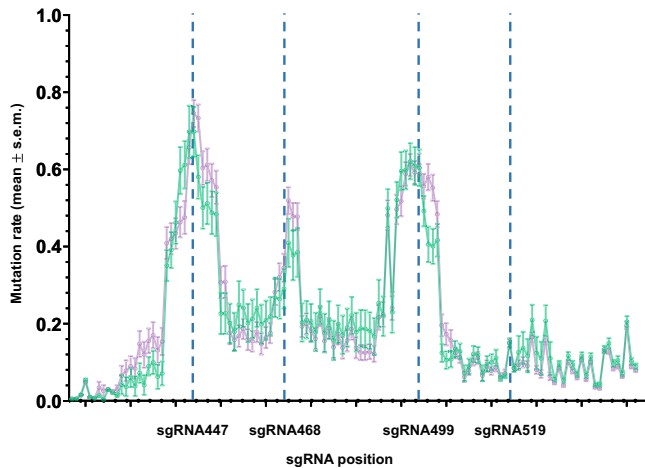

**Fig. 3 | Mutational rates vary for *kmo* targets sgRNA447, sgRNA468, sgRNA499, and sgRNA519.** Dashed blue lines represent the expected cut sites of corresponding sgRNAs. Green line represents white-eyed $F_2$ larvae which did not inherit the *kmo*$^{sgRNAs}$ transgene (non-transgenic or inheriting only the *bgcn*-Cas9 element) from the ♂ *kmo*$^{sgRNAs}$; *bgcn*-Cas9 x ♀ *kmo*$^{-/-}$ cross. Purple line represents reciprocal cross: white-eyed $F_2$ larvae which did not inherit the *kmo*$^{sgRNAs}$ transgene from the ♂ *kmo*$^{-/-}$ x ♀ *kmo*$^{sgRNAs}$; *bgcn*-Cas9 cross. Data points are the mean proportion of mutant nucleotides at each position of the *kmo* region targeted as determined by CRISPResso2, error bars are the s.e.m. (standard error of the mean). Source data are provided as a Source Data file.

females show significantly higher rates of cutting compared to the offspring of males (Binomial GLMM: Odds ratio = 10.47 [2.16–50.78], $p = 0.004$, Table S9 Model 4), possibly due to additional cutting from maternally deposited nuclease activity, however this may also be reflective of different germline activity between the sexes.

White-eyed $F_2$ larvae which did not inherit the *kmo*$^{sgRNAs}$ were collected for deep sequencing to determine the relative cutting efficiency of each sgRNA (Fig. 3). It is important to note that the mutations observed in these individuals include mutations originally present in and contributed by the *kmo*$^{-/-}$ line (Fig. S5). Results from this deep sequencing can therefore only indicate the relative frequency of mutations caused by respective sgRNAs but do not indicate the timing at which nuclease activity occurred or if cuts by a certain sgRNA bias HDR over NHEJ upon cleavage. We determined the prevalence of mutated nucleotides in the sequence reads relative to the wild type *kmo* sequence. We found a wide range of cleavage events for each unique pol III expressed sgRNA. sgRNAs 447 (U6.763) and 499 (U6.702) seem to be the most active sgRNAs resulting in approximately 61–76% of the alleles cut. As the target sites slightly overlap, mutations at the target site of sgRNA 447 may alter part of the 5' end of the sgRNA468 target making it unable to cleave (Fig. 1a). sgRNA 468 (Ae7SK) had ~50% of alleles cut and for 519 (U6.774) a mere 15–16% of alleles were mutated (Fig. 3). Simultaneous cuts between the two outermost sgRNAs (447 and 519) would generate deletions of 72 nt that eliminate all four sgRNA targets and create a fully cut-resistant allele. We did not observe any such deletions among the non-*kmo*$^{sgRNAs}$ inheriting larvae collected (Fig. S6). Deletions which span between two target sites were observed, however, the majority of indels appear to be the result of single cuts. Much larger deletions that could remove one or both primer binding sites would not be readily distinguishable by this assay. This may be the case with sample CUT3WT, which has three different deletions adding up to 43 nt (and 3 bp substitution) which appears to be homozygous. As this mutation is not present in the *kmo*$^{-/-}$ line the most likely scenario is that one or both primer binding sites are missing from the other allele, such that only one allele was amplified and sequenced. This sample is also the only sample which has deletions encompassing part of all four sgRNA target sites,

although for sgRNA target 499 only the five most 5' nucleotides are affected. As this is most likely only one allele it is unclear whether further cleavage could occur in this individual, and the white eyed phenotype indicates this is an r2 mutation, not a functional allele. Having characterized the isolated metrics of the drive, we next set out to test its performance at the population level.

## Cage trials

To evaluate the ability of the split-drive design to spread through a WT population, we initiated two replicate experimental cages (A1 and A2) by mating 100 mosaic eyed female *kmo*$^{sgRNAs}$;*bgcn*-Cas9 trans-heterozygotes to 100 wild type males ($F_0$) and monitored both transgenes as well as eye phenotypes for six generations (Fig. 4b, Dataset S1). Although such a population set up may not be realistic of the potential use of such a system in the field, it was chosen to allow robust data to be collected on the dynamics of spread of this proof-of-principle system in a reasonable time-frame. In the $F_1$ generation we observed an increase in the proportion of the population carrying the *kmo*$^{sgRNAs}$ element followed by a small decrease in the $F_2$ generation (79% to 76.5% in A1 and 77.4% to 75% in A2). In the $F_3$ generation the frequency of the *kmo*$^{sgRNAs}$ substantially diverges between the replicate cages (74% in A1 and 88% in A2), presumably due to stochastic effects, but still remains within the model-predicted range. We observed a maximum *kmo*$^{sgRNAs}$ frequency of 89% in these small cage populations, consistent with the upper end of the stochastic model prediction (Fig. 4b, Dataset S1).

By also noting the eye color phenotype through the generations of the cage trial we can gain insight into NHEJ rates in individuals which do not carry either element. A mosaic eyed phenotype we take to be an indication of embryonic deposition of drive components, or somatic expression if the individual has both transgenes. For individuals carrying the *kmo*$^{sgRNAs}$ transgene the mosaic eye phenotype frequency decreased from the 100% in the initial trans-heterozygotes ($F_0$) to about 70% in the $F_1$ generation and stayed below 10% thereafter, as was similarly observed in our initial test crosses (Table S3, S5). In those which did not carry the *kmo*$^{sgRNAs}$ transgene, mosaicism was about 25% in the second generation. This is similar to rates we observed in the test crosses (Table S8). However, from then on, no mosaicism was observed in non-*kmo*$^{sgRNAs}$ individuals, except for a small number of individuals in the fourth generation in cage A1 (Dataset S1).

In the experimental cages, a complete white eyed phenotype indicates that both *kmo* alleles are disrupted. In those mosquitoes which carry the *kmo*$^{sgRNAs}$ element, we could observe white eyes in individuals either homozygous for the *kmo*$^{sgRNAs}$ element or heterozygous for this element with the other *kmo* allele disrupted by a non-functional mutation. We observed an increase in the frequency of white eyes in *kmo*$^{sgRNAs}$ mosquitoes reaching a maximum of 89.15% (A1) and 82.44% (A2) in the third generation (Fig. S9, Dataset S1). In mosquitoes which do not carry the *kmo*$^{sgRNAs}$ element the presence of white eyes corresponds to disruption of both *kmo* alleles. In those mosquitoes which did not carry the *kmo*$^{sgRNAs}$ element, a maximum of 60% were observed with white eyes.

## Modeling of drive behavior

Fitness effects of the two transgenic constructs used in this study were explored using a deterministic, discrete generation, population genetics mathematical model. A stochastic modeling framework[23] was also used to provide a prediction as to the potential range within which we would expect experimental results to vary. We begin with a simple model describing the behavior of a single transgenic construct (i.e., in absence of homing) and use a simple least-squares regression approach to obtain fitness parameters for heterozygous and homozygous individuals. Full details of the deterministic and stochastic mathematical models and parameter fitting procedure are given in Supplemental information S2. Briefly, for *bgcn*-Cas9 the best fit of the deterministic model to the experimental data is obtained where

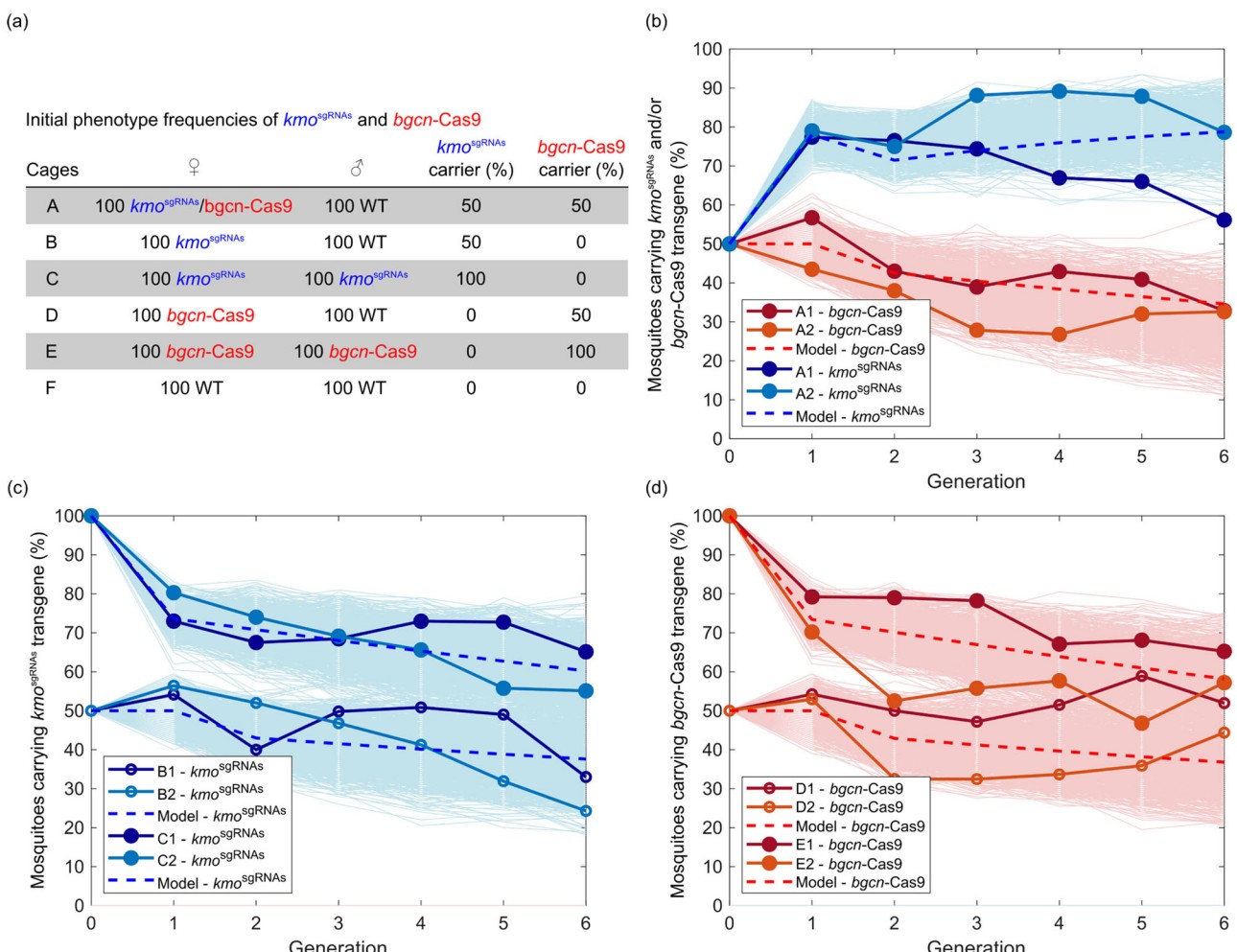

**Fig. 4 | *bgcn*-Cas9; *kmo*^sgRNAs split-drive can increase in frequency through a caged population.** Presence of *kmo*^sgRNAs and *bgcn*-Cas9 elements observed for 6 generations of small, caged populations and the percentages predicted by the deterministic (dashed lines) and stochastic (pale lines) models. Cages each generation beyond the initial setup were begun with 250 L1 larvae. **a** Initial frequencies for each cage at the outset of the trial. **b** Percentage of individuals carrying the *kmo*^sgRNAs transgene (blue solid lines) and/or the *bgcn*-Cas9 transgene (red solid lines). **c** Percentage of *kmo*^sgRNAs transgene in the absence of *bgcn*-Cas9 when female

heterozygotes are crossed to either male heterozygotes (filled circles) or to wild type (WT) males (open circles). **d** Percentage of *bgcn*-Cas9 transgene in the absence of the *kmo*^sgRNAs, again with female heterozygotes crossed with male heterozygotes (filled circles) or to WT males (circles). The stochastic results show the behavior produced by each of 1000 independent numerical simulations. F cages were used to estimate the population size at each generation, details are included in Supplementary Dataset S1 and source data file.

heterozygous and wild type individuals are equally fit while homozygotes have a fitness cost of 21% (S4 Fig. S1, with model output in Fig. 4d). Non-exclusive potential explanations of such a fitness cost could be, for example, a deleterious threshold of Cas9 expression, insertional mutagenesis at the target site, or insertion linked to deleterious recessive alleles[42]. Using the same approach, we obtain a best fit for *kmo*^sgRNAs where heterozygotes have wild type fitness and homozygous individuals have a fitness cost of 19% (S4 Fig. S2, with model output in Fig. 4c). There is contradictory evidence relating to fitness costs of *kmo* in the literature, most notably a high load observed in *An. stephensi*, although knock-outs and knock-ins were previously described in *Ae. aegypti* fitness effects were not measured[18,39]. In our own recent experience with *Culex quinquefasciatus kmo*^-/- could be generated and maintained as homozygotes, but an insertional mutant expressing a fluorescent protein was homozygous lethal[43,44]. Using these best fit parameter values within the stochastic model shows that experimental results fall within the expected range. While these parameters produced the best fit of the deterministic model to experimental data, Figs. S1a and S2a from S2 demonstrate a range of relative fitness parameters that can produce a similarly good fit.

We then utilize a deterministic population genetics mathematical model including both transgenic constructs and the effect of inheritance bias to predict the behavior observed within the experimental treatment cages (full details are available in S2). This model was parameterized using directly measured inheritance rates (Fig. 2) and the fitness parameters obtained above. For the remaining genotypes (i.e., those carrying both constructs) additive and multiplicative combinations as well as independent least-squares regression for the fitness of each genotype were compared. Each approach yielded only a marginal difference in the goodness of fit. We therefore considered additive parameter combinations since they provide a simple and intuitive explanation of interactions between multiple fitness parameters. These were used to predict the behavior of the split-drive system using both deterministic and stochastic mathematical models, giving a fit to experimental data that is broadly within the expected range (Fig. 4b). This suggests that our assumed model of the drive behavior and all parameters derived here provide a good understanding of the system (at least in our cage trial setting). We found some minor differences between the model predictions and the experimental data that are likely caused by factors not considered within these mathematical

models (e.g., multiplexed sgRNAs, end-joining mediated resistance and maternal deposition of transgenic constructs). Note that changes in the availability of intact target sgRNA target sites have been neglected within the mathematical model due to the relatively short timescale considered - which is supported by the broad agreement between the models and experimental data. However, we would expect this to be of great importance when modeling the efficacy of gene drive systems that persist over longer timescales. We also note that fluctuations due to stochastic effects appear larger in the experimental results than in the results of our models. This is potentially due to the effective population size being lower than the census size of the caged populations, with the latter being used within our models.

**Multiple sgRNA recognition sites remained intact after the fifth generation of the trial**

Having determined the mutant alleles generated from single generational crosses (Fig. S6), we investigated the types of mutant alleles that were formed in multiple generations through the cage trial. We collected mosaic and white eyed individuals from the experimental cages (A1 and A2) at generations $F_2$, $F_4$, and $F_5$ for deep sequencing (Fig. S9, Table S10). We predict that alleles which are cut-resistant at multiple target sites are likely to present with a null phenotype. The proportion of WT sequence was calculated using CRISPResso2 and plotted (Fig. 5). In this diagram a higher percentage (*y*-axis) indicates a greater abundance of unmodified nucleotides from the samples. Three replicates of five wild type adults each were also analyzed to assess the prevalence of naturally occurring SNPs that are present in our wild type population.

Analysis of the sequence reads of wild type samples showed that all nucleotides within the analysis window were identical [>99.7% of reads showed wild-type sequence] to the reference sequence (Sanger sequencing of the *kmo* allele from a single individual collected from our wild-type population) against which they were aligned (Fig. 5). In the mosaic- and white-eyed individuals collected from the cage trial, wild type reads surrounding the cut sites were reduced but they did not exhibit a pattern of continuous decline from generation $F_2$ to $F_5$, indicating the absence of any substantial accumulation of mutant alleles between these generations. In particular, an average of at least 60.9% and 80.3% of nucleotides on the recognition sites for sgRNAs 468 and 519, respectively, are still unmodified by generation $F_5$ (Fig. 5). Separately, we also collected and analysed wild type-eyed, *kmo*^sgRNAs-inheriting larvae from the same generations of the cage trial to investigate the potential r1 mutations that may have formed and/or accumulated (Fig. S9a). Since they inherit the *kmo*^sgRNAs element which is a null allele, any singular (i.e. not mosaic) mutations of the *kmo* allele on the homologous chromosome would be an r1 mutation. In the later generations we identified several individuals with a 3-bp deletion in the target site of our most active sgRNA, 447 (Fig. S11). We do not however, know the fitness of these mutants compared to wild type or null alleles and therefore cannot predict their behavior in a population over time.

Taken together, we have shown that among the mosaic- and white-eyed larvae, recognition sites for at least two sgRNAs are still largely available for further cuts to occur and that even the potential r1 mutations that may have formed during the first five generations of the cage trial had at least one sgRNA recognition site still intact.

## Discussion

Multi-generational lab trials are a critical step towards assessing the utility of novel gene drive systems in the field by considering complex fitness components such as fecundity, longevity, and mating competition. Here we evaluated the spread of a CRISPR/Cas9 multiplexing split-drive in multi-generational, caged lab populations of *Ae. aegypti*. Using regulatory sequences from the *Ae. aegypti bgcn* homologue to express Cas9 in the germline and a separate sgRNAs expressing cassette integrated into the *Ae. aegypti kmo* gene; we demonstrated highly

effective germline cutting rates and bias in the inheritance of our genetic element. The frequency of individuals carrying at least one allele increased from an initial 50% to a maximum of 89% in five generations, in line with the upper bound predicted by stochastic modeling. These results showed an improvement in the inheritance bias in this mosquito species compared to previous studies[38]. Throughout our cage trial, the drive produced substantial increases in cut-resistant alleles across all four target sites, but no deletions which removed all four target sites. While potentially functional mutant alleles did arise in the later generations of the cage trial, deep sequencing of representative individuals revealed that sgRNA target sites were still intact and could still be targeted by the drive. Interestingly, the target sites most available in these individuals (519 and to a lesser degree 468) were the most active as determined in a previous study[39]. Additionally, these were paired with pol III promoters which were believed to be highly active from previous works either in vivo or in vitro[37,38]. It is not wholly clear why these sites remained uncut; it may be that there is some degree of position effect on the expression of pol III promoters. This also may be due to the timing of expression from the different pol III promoters used. Expression outside of the timing which homing can occur could favor NHEJ, and this would be reflected in the mutations that we observed.

A complementary strategy to managing target-site sequence variation has targeted highly conserved and, ideally, functionally constrained sequences with a single sgRNA[10,45]. This strategy has proved highly effective and combining these approaches would likely improve gene drive conversion efficiencies through further reduction of resistance allele formation, although new designs may be required as the highly conserved RNA sequences used to date are likely too small to allow the identification of multiple sgRNA targets. More complex strategies such as targeting and recoding essential genes could also be used, which should provide selection against r2 alleles and allow the target sgRNA expressing allele to approach fixation[21,32,36,46].

Model fitting to the cage trial data showed that both elements of the split-drive carried moderate fitness costs expressed in homozygotes which may have impeded the rate of spread in the drive and prevented the drive from reaching fixation. We found a significant difference in observed mean inheritance rates in the offspring of males between pooled and individual mating crosses. This observed difference was likely due to mating competitiveness, as in individual crosses all males have an equal chance to mate, but in pooled crosses some genotypes may contribute fewer offspring to the next generation. We also found high variability in the apparent inheritance bias and cutting rates between individuals of both sexes in our individual mating crosses. This likely represents several different fitness costs at work. Taking into consideration batch effects with individual crosses, we found the inheritance bias to be the same between male and female $F_1$. The insertion sites of the transgenes in this study played an essential role in Cas9 expression and efficacy, and we found high recessive fitness costs in our selected *bgcn*-Cas9 transgenic. Analysis of additional insertion sites or use of insertions into the endogenous locus could yield lines which maintain the ability to bias inheritance or even improve upon it, with lower associated fitness costs. Similarly, *kmo*^sgRNAs also showed significant recessive fitness costs, which was not anticipated when this program was initiated. To maximize the efficiency of a "population modification" drive, it will be essential to identify "cargo" elements and drive insertion sites that minimize fitness costs.

The control of the expression of Cas9 in gene drive systems is critical, as expression either too late in the germline or in somatic cells is likely to result in repair by NHEJ and the formation of cut-resistant alleles[36,47]. *bgcn* has been identified and characterized as a regulator of cystoblast formation in *D. melanogaster*. Transcripts are restricted to a few cells, including germline stem cells. This pattern should be ideal

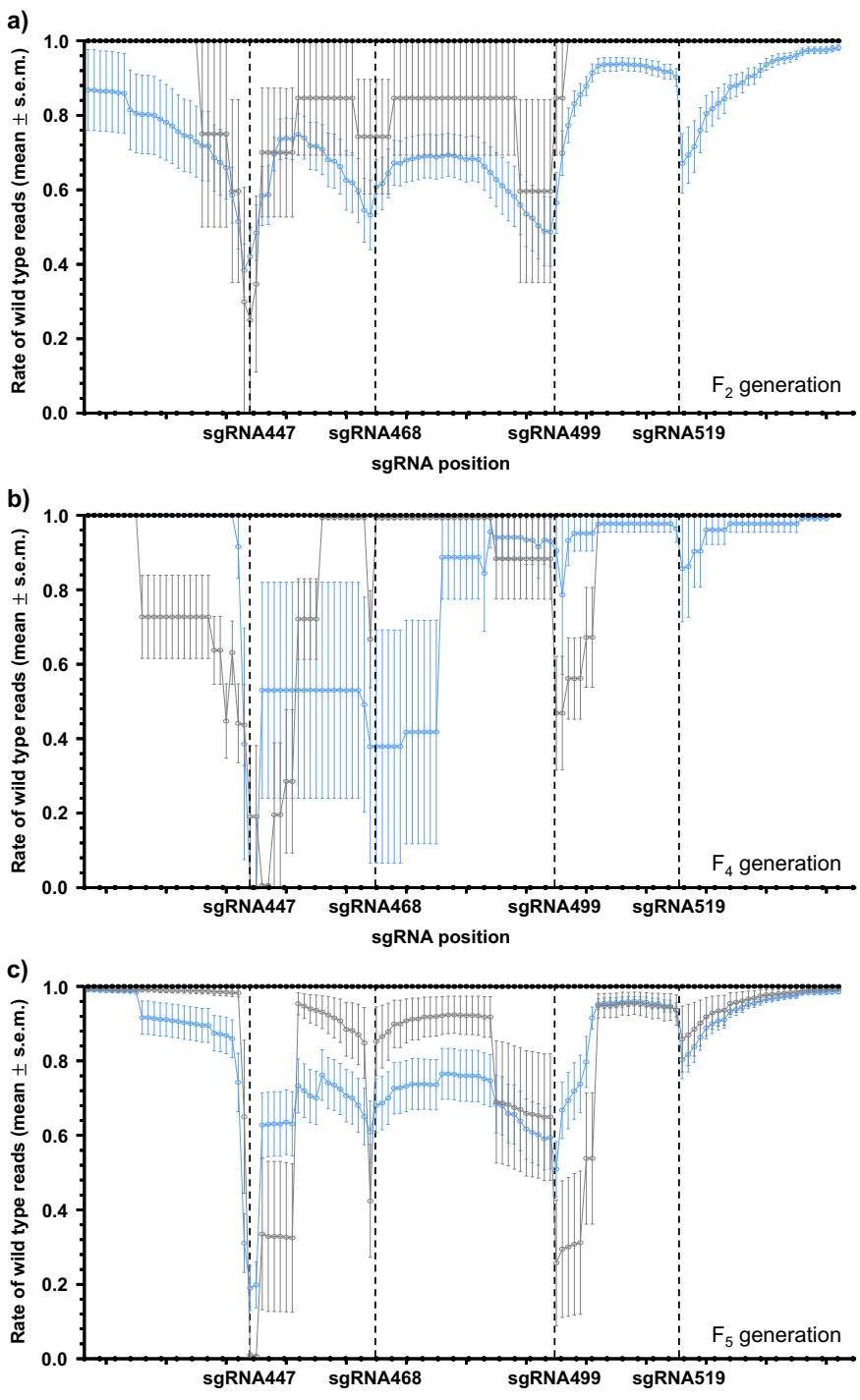

**Fig. 5 | Individuals collected from the cage trial have wild-type *kmo* alleles.** Liverpool (WT) samples are represented by black circles while *kmo*^sgRNAs^-inheriting and non-*kmo*^sgRNAs^-inheriting individuals from F$_2$ (**a**), F$_4$ (**b**) and F$_5$ (**c**) generations of the cage trial are represented by blue and grey hollow circles, respectively. Data points are the mean and error bars are the standard error for the mean (s.e.m.) of the proportion of wild-type nucleotides at each base of the *kmo* region targeted as determined by CRISPResso2. Source data are provided as a Source Data file.

for confining Cas9 expression to the germline and minimizing mosaicism. Our results, however, indicate some somatic expression which means that either our transgene could not recapitulate the endogenous expression pattern of this gene and/or there are significant differences in the expression pattern between *D. melanogaster* and *Ae. aegypti*. In publicly available *Ae. aegypti* RNA-Seq datasets *bgcn* was found to be expressed in females in ovaries both pre- and post-bloodmeal as well as male and female brains but more precise

localization data are not available[48–50]. There is clear scope for the identification of further germline specific genes which can be used either in the endogenous context or whose regulatory elements could be used to express nucleases from a transgene construct such as the recently reported *shu* or *sds3*[51]. The observed differences in inheritance bias between males from pooled and individual crosses may have captured the effect of small fitness loads in the heterozygotes (or those which are strong mosaics and thus somatically homozygous) on male

sexual competitiveness. In pooled crosses, males with minimal transgene expression may gain disproportionate shares of reproduction, though the underlying mechanism for fitness costs is unknown. Previous work in this system has also noted similarly high levels of individual variation to those we observed in our study of inheritance bias rates across both sexes[38]. One of the strengths of split gene-drive systems is that it allows future work to test new constructs in different combinations, which would allow these issues to be addressed in the future.

Maternal deposition of Cas9/sgRNAs in our drive may have acted to increase the inheritance rates of the *kmo*[sgRNAs] transgene rather than resulting in NHEJ, perhaps due to the multiplex design. In split-drives using a single sgRNA target, maternal deposition often resulted in early embryonic cutting favoring NHEJ rather than HDR, generating resistance alleles at the expense of homing[29,32,41,52]. With additional target sites still available in our design there may be additional opportunities for deposited Cas9 to cleave within a later HDR-conducive window, resulting in some level of transgenerational effect ("shadow drive")[52,53]. Further studies into additional germline specific promoters could improve inheritance rates and decrease NHEJ resulting from somatic expression and/or deposition of Cas9. Improved germline specificity could be optimized through promotor selection as well as other methods for restricting transcript and/or protein. Improvement to cutting rates has already been demonstrated with newly characterized germline promoter sequences[51]. Taking these factors into account, optimizing sgRNA efficiency and pol III promoter expression a CRISPR/Cas9 gene drive appears feasible in *Aedes aegypti*.

## Methods

### Plasmids and cloning

**Design and cloning of *kmo*[sgRNAs] multiplexed sgRNA expression construct.** To generate the *kmo*[sgRNAs] knock-in plasmid we first sought to sequence confirm the *kmo* locus of our Liverpool strain. Those regions upstream and downstream of exon 5 which we were able to confirm were used as homology arms (1942 bp upstream and 1241 bp downstream of our target sites) in the final construct. An Hr5/IE1 AmCyan K10 3'UTR cassette (AGG1036) was used to enable detection of the transgene by fluorescent microscopy. The multiplexed sgRNA expression cassette was synthesized (Genewiz) to contain an array of four cassettes each consisting of 600 bp upstream region of an endogenous *Ae. aegypti* pol III RNA (Ae U6.763 (AAEL017763), Ae U6.774 (AAEL017774), Ae U6.702 (AAEL017702), Ae 7SK (AAEL018514))[37], an sgRNA targeting exon 5 of the *kmo* gene (cutting at 447, 468, 499, and 519 bp into exon 5 of *kmo*)[39], with one of four sgRNA backbone variants (23 with a 5 bp extended stem loop, 29, 9, 25)[40], and a poly-T (7 nt) terminator for the pol III promoter (Figs. 1, S1). Three of these targets are previously validated[39] and the fourth was designed using CHOPCHOP and selected by location, closest off-targets for all sgRNAs as determined by CHOPCHOP are listed in Table S12. Complete primer sequences are listed in Table S11. Plasmid sequence is available through NCBI accession number: OP728003 {https://www.ncbi.nlm.nih.gov/nuccore/OP728003}.

### Identification of germline promoter, design and construction of *bgcn*-Cas9 expression plasmid

Blastp using the *D. melanogaster* amino acid sequence was performed. The *Ae. aegypti* ortholog was identified (with 28% aa sequence identity) as AAEL004117, annotated as an ATP-dependent RNA helicase, consistent with the *D. melanogaster* gene annotation. The *bgcn*-Cas9 expression construct was built based on plasmids kindly provided by Omar Akbari, with several modifications[38] (Figs. 1b, S1). The fluorescent marker OpIE2-DsRED cassette was replaced with more easily visualized *AePUb*-mCherry[53]. The human codon optimized *Streptococcus pyogenes*-Cas9 was replaced with an insect codon optimized version (VectorNTI) synthesized by Genewiz. This was generated using the

Regenerator tool in VectorNTI using the *Aedes aegypti* codon usage table, we then scanned for rare codons in *Plutella xylostella* and *Anopheles gambiae* and manually changed them so that they were not rare for any species, we then checked for cryptic splicing using the Berkeley Drosophila Genome Project splice site prediction (https://www.fruitfly.org/seq_tools/splice.html) and modified any strong splice sites manually. 5' and 3' RACE ready cDNA was generated from RNA extracted from ~30 pairs of ovaries or testes dissected from 5–7 days-post-emergence (dpe) Liverpool strain adults using Trizol (Invitrogen). Primers LA1076 then nested with LA1352 (5') and LA1074 then nested LA1075 (3') were used to amplify the 5' and 3' ends of the cDNA transcript and these amplicons were sequenced to verify the annotated UTRs (Fig. S7, Table S11).

*Aebgcn* promoter and 3'UTR fragments were amplified using primers LA1725 and LA1726 (2213 bp upstream of ATG) and LA1737 and LA1738 (629 bp downstream of stop codon) (Table S11) from genomic DNA prepared from our Liverpool WT colony using the NucleoSpin Tissue kit (Macherey-Nagel) and ligated into the plasmid sequentially by standard restriction enzyme-based cloning to generate *bgcn*-Cas9 (AGG1207). Plasmid sequence is available through NCBI accession number: OP728005. Complete primer sequences are listed in Table S11.

### An improved *piggyBac* helper plasmid

Hyperactive *piggyBac*[54] has been used to increase the insertion efficiency in insects and so we synthesized (Genewiz) an *Ae. aegypti* codon optimized (ATGme)[55] version. This along with pGL3-PUb (gift from Zach Adelman, Addgene plasmid # 52891; http://n2t.net/addgene:52891; RRID:Addgene_52891) were digested with *Nco* I and *Fse* I and ligated using T4 DNA ligase (NEB M0202S) to generate AGG1245. Plasmid sequence is available through NCBI accession number: OP728004.

### Mosquitoes, transgenics and cage trial

All experiments performed for this study were reviewed and approved by the Biological Agents and Genetic Modification Safety Committee (BAGMSC) at The Pirbright Institute.

### Mosquito rearing

*Ae. aegypti* Liverpool strain (WT) was used for all experiments. All mosquitoes were reared under constant conditions: 28 °C, 65–75% relative humidity and 14:10 light/dark cycle with 1 h of dawn and 1 h of dusk. Larvae were fed with ground TetraMin flake fish food (TetraMin 769939) and adults were provided with 10% sucrose solution *ad libitum*. Females were blood fed with defibrinated horse blood (TCS Bioscience HB030) using a Hemotek feeder (Hemotek, Inc AS6W1-3) covered with Parafilm (Bemis HS234526B).

### Microinjections, crosses, screening

Transgenic *Ae. aegypti* mosquitoes were generated by microinjection of embryos less than 2 h post oviposition as described previously[56]. In brief, 1 h embryos were collected, and aligned using a fine paint brush. Lines of ~100 embryos were transferred to double-stick tape and covered in Halocarbon oil 27 (Sigma H8773) after a few seconds of desiccation. Needles were generated by pulling Quartz capillaries (Sutter QF1007010) using a P2000 laser pipette puller (Sutter). G$_0$ embryos were hatched one week after injection and larvae reared as described above. For the generation of the Cas9 line, embryos were injected with 500 ng/µl AGG1207 and 300 ng/µl AGG1245 (PUb hyperactive *piggyBac* transposase) in 1X injection buffer. For generation of *kmo*[sgRNAs] transgenics, embryos were injected with 300 ng/µl Cas9 protein (PNABio CP01), in vitro transcribed sgRNAs at 40 ng/µl sgRNA447, 40 ng/µl sgRNA519, and 300 ng/µl AGG1095 in 1x injection buffer.

Templates for in vitro transcription were designed as described previously[57] using overlapping oligos and extending by PCR with

LA925 (sgRNA447), LA926 (sgRNA519), and LA924 (common R) (Table S10). sgRNAs were in vitro transcribed using the MEGAscript T7 in vitro transcription kit (ThermoFisher AM1333) according to the manufacturers' instructions. RNA was purified using the MEGAclear in vitro transcription reaction clean-up kit (ThermoFisher AM1908) aliquoted and stored at −80 °C until use. Complete primer sequences are listed in Table S11.

All $G_0$ adults were crossed to WT mosquitoes. $G_0$ males were crossed individually to 5 WT females for 2–3 days and then pooled to approximately 20 $G_0$ individuals in a cage, while $G_0$ females were crossed to WT males as a pool of approximately 20 $G_0$ females to 20 WT males. $G_1$ progeny were screened for presence of the fluorescent marker using a Leica MZ165FC microscope (Leica Biosystems).

### Generation of white eyed mutant ($kmo^{-/-}$) strain

To determine the rate of CRISPR/Cas9 induced cutting and germline inheritance bias, a $kmo^{-/-}$ knockout line was generated by crossing two white eyed non-drive inheriting individuals (one male and one female) generated from an inheritance assessment cross. The region encompassing the sgRNA recognition sites was amplified with primers LA1275 + LA518 and mutations identified by Sanger sequencing (Eurofins) and listed in Fig. S5. Deep sequencing of four replicates of $kmo^{-/-}$ adults ($n = 24$) indicates there are at least eight distinct $kmo$ knockout alleles in the $kmo^{-/-}$ line even though the line was generated by crossing only two non-drive-inheriting founders (Fig. S5). It is likely that the different mutant alleles in the germline of the founders were generated by nuclease activity originating from (i.e., deposited by) their trans-heterozygous parent. Complete primer sequences are listed in Table S11.

### Confirmation of insertion

Adapter-ligation mediated PCRs (Supplementary text S1) were performed on $bgcn$-Cas9 transgenic lines according to previously reported methods[58,59]. gDNA was extracted from 10 individuals from $bgcn$-Cas9 using the NucleoSpin Tissue kit (Macherey-Nagel 740952.50). DNA was digested with the restriction enzymes BamHI (NEB R3136), MspI (NEB R0106) and NcoI-HF (NEB R3193) and PCRs were performed with DreamTaq (Thermo Fisher Scientific EP0712) and primers LA182, LA184, LA186 and LA187. Complete primer sequences are listed in Table S11.

Genomic DNA was extracted from a single founder $G_1$ male using the NucleoSpin Tissue kit (Macherey-Nagel 740952.50) and subjected to two separate PCR reactions with primers LA2750, LA174 and LA1301, LA2755 to confirm correct homology-directed repair of the construct (Fig. S4). PCR amplicons although they appear larger than expected were further sequence confirmed by Sanger sequencing and the junction between the homology arms and the genome was confirmed. It is likely that the size discrepancy is due to variability in the introns which are included in this amplicon. Complete primer sequences are listed in Table S11.

### Phenotype data analysis

$kmo^{sgRNAs}$ adult females and males (at least 20) were crossed to the opposite sex $bgcn$-Cas9 adults to generate trans-heterozygous adults ($F_1$). For initial assessments of $kmo^{sgRNAs}$ transgene inheritance, $F_1$ adults were pooled into groups of at least 5 transheterozygous females or males and crossed to WT. All $F_1$ trans-heterozygotes displayed a mosaic eyed phenotype. Progeny ($F_2$) were screened for presence of each transgene and eye color phenotype (Fig. 2a). We used a likelihood ratio test to compare rates of transgene inheritance compared to an expected distribution under standard Mendelian inheritance. We were able to interrogate additivity and total fit to compare the effects of insertion site, maternal/paternal inheritance of Cas9, mosaicism and replication on assessments of inheritance bias. Exponentiated log odds and standard errors were used to generate approximate 95% confidence intervals. This pooling approach does not take into account

potential individual differences in fitness, mating rates or inheritance bias. A replicate cross was then performed, again starting with female $bgcn$-Cas9 crossed to $kmo^{sgRNAs}$ males, the $F_1$ transheterozygous females ($n = 65$) were crossed to WT, bloodfed, then allowed to lay individually and the $F_2$ progeny hatched and scored for inheritance of the $kmo^{sgRNAs}$ transgene (as indicated by AmCyan fluorescence) and eye phenotype (Fig. 2b).

To accurately quantify rates of Cas9 cleavage in relation to inheritance bias, $F_1$ trans-heterozygous females and males generated by crossing $kmo^{sgRNAs}$ males to $bgcn$-Cas9 females were also crossed to a $kmo^{-/-}$ line. Crosses were performed as single pair crosses, and females were allowed to lay eggs individually. Progeny ($F_2$) were screened as before for the presence of each transgene and eye color phenotype. Analyses for the proportion of $F_2$ progeny with white eyes and $kmo^{sgRNAs}$ inheritance were made by fitting a generalized linear mixed model, with a binomial ('logit' link) error distribution. This accounts for replication, and results in slightly different estimates from pooled data, with increased estimate intervals.

White-eyed progeny without the transgene were snap-frozen in liquid nitrogen and stored at −80 °C for further analysis. Genomic DNA was extracted using the NucleoSpin Tissue kit (Macherey-Nagel 740952.50). Further sequencing was carried out by Illumina MiSeq using primers LA4507, LA4508 flanking a 500 bp fragment including the sgRNA target sites following a previously published procedure and detailed below[60]. Complete primer sequences are listed in Table S11.

### Cage trial

A cage trial was undertaken to study the performance of the split $bgcn$-Cas9;$kmo^{sgRNAs}$ drive in a small laboratory population. A total of 12 cage populations were established (6 ratios in duplicate, designated 1 and 2 for each condition) with the following adults: experimental cages 100 $kmo^{sgRNAs}$;$bgcn$-Cas9 trans-heterozygous females and 100 WT males (A1 and A2); control cages of 100 $kmo^{sgRNAs}$ heterozygous females and 100 WT males (B1 and B2), 100 $kmo^{sgRNAs}$ heterozygous females and 100 $kmo^{sgRNAs}$ heterozygous males (C1 and C3), 100 heterozygous $bgcn$-Cas9 females and 100 WT males (D1 and D2), 100 heterozygous $bgcn$-Cas9 females and 100 heterozygous $bgcn$-Cas9 males (E1 and E2), and 100 WT females and 100 WT males (F1 and F2) (Fig. 4a). Individuals destined for cages A, B, and C were derived from an initial cross of trans-heterozygous ($kmo^{sgRNAs}$; $bgcn$-Cas9) males to WT females. Adults for $bgcn$-Cas9 only cages (D and E) were selected from a maintenance $bgcn$-Cas9 line (generation 9). WT adults for cages F were selected from the Liverpool mosquito line. The trans-heterozygous females used to establish cages A1-2 presented mosaic eyes, so the initial frequency for this eye phenotype in the experimental cages was 50%. To establish each generation of the cages, eggs were hatched in degassed reverse osmosis water and 250 L1 larvae/condition were randomly separated using a Biosorter (Union Biometrica). To keep all conditions as homogeneous as possible, larvae were reared in standardized trays with a set volume of water (2L) and following a feeding scheme described previously in[61]. Pupae were separated by sex and females and males allowed to eclose separately in cages and provided with 10% sucrose *ad libitum*. Five days post eclosion all adults were anesthetized with $CO_2$ and simultaneously transferred to the final cage (W24.5 × D24.5 × H24.5 cm) (BugDorm 4M2222) so all the adults would have the same chances to mate. The trial was continued for six generations (Fig. S8). At each generation, two ovipositions were collected from each cage and after the second oviposition the adults were snap-frozen and stored at −80 °C for further molecular analysis. Screening for fluorescence and eye phenotype were performed at pupae stage.

### Amplicon sequencing

Amplicon sequencing was carried out as previously published[60]. gDNAs were extracted using the NucleoSpin Tissue kit (Macherey-Nagel 740952.50). Approximately 500 bp surrounding the sgRNA

target sites was amplified using primers listed in Table S11. A second round of PCR was performed using the Nextera XT index kit, and Nextera XT index kit D (Illumina FC-131-1001 and FC-131-2004). Amplicon sizes were verified on a Tapestation using the High Sensitivity D1000 Screentape (Agilent 5067-5584). The NEBNext Library Quant kit (NEB E7630L) was used to quantify the amplicons prior to pooling. Sequencing was carried out by the Bioinformatics, Sequencing and Proteomics facility at The Pirbright Institute.

### Reporting summary

Further information on research design is available in the Nature Portfolio Reporting Summary linked to this article.

## Data availability

All raw reads from amplicon-Seq generated in this study were submitted to NCBI SRA with the accession number PRJNA741076. Addgene plasmid # 52891; http://n2t.net/addgene:52891; RRID:Addgene_52891. Plasmid sequences are available from NCBI OP728003 https://www.ncbi.nlm.nih.gov/nuccore/OP728003, OP728004, OP728005 https://www.ncbi.nlm.nih.gov/nuccore/OP728005. The remaining data generated for this study is available in the Supplementary dataset 1 file. Source data are provided with this paper.

## Code availability

**Amplicon-Seq analysis** Sequencing reads generated from the amplicon sequencing were analysed using the CRISPRessoBatch tool in CRISPResso2 ([62] with the following script: *CRISPRessoBatch --batch-settings[**batch file name**] -a ttatgatgatcgccctgcccaatcaggatcgcacttggac ggtgacgctgttcatgccgttcaccaacttcaacagtattaagtgcgatggcgatttgttgaagttc ttccggacatacttccccgatgcgattgatctgattggtcgtgagcggttggttaaggatttctttaa gaccaggcctcaatcgttggttatgatcaagtgtaagccatataatgtgggcggcaaggcggtg atcattggtgatgcggcacatgccatggttcccttctacgggcagggaatgaatgccggattcga ggaTTGTACTGTGTTGACCGAGTTGTTCAATCAACATGGCAGTGACGTTG ATAGGATACTGGCTGAGTTTAGTGATACGCGTTGGGAGGATGCACACTC TATCTGCGATCTGGCCATGTATAATTATGTTGAGGTTAGTATATGGTCTT TTATTTATATCGTACGTTTTGTATGCGGTCGTTTTGTAGGTACCGTA -g gc catatataatgtgggcggca,ggcggtgatcattggtgatg,ggttcccttctacgggca,CACAGT ACAAtcctcgaatc -q[**20 or 30**] -qwc 211-254_264-285_294-316 --offset_around_cut_to_plot 80 --skip_failed* Modification rates of nucleotides surrounding the sgRNA recognition sites were plotted with GraphPad Prism 9. In cases of insertions, CRISPResso2 counts the nucleotides on both sides of the insertion as mutant in the output file. In this case rates were calculated using only the 'insertion left' dataset, to avoid counting the same mutation twice. Rates of unmodified nucleotides were calculated by simple subtraction (1 - modification rate) and subsequently plotted with GraphPad Prism 9. **Phenotype data analysis** We carried out all phenotype analyses using R version 3.6.2 (R Development Core Team)[61]. Data sets were summarized using 'tidyverse'[63] and figures generated using 'ggplot2'[64]. Likelihood ratio tests carried out with 'DescTools'[65]. Generalized linear mixed model analyses were carried out using 'lme4'[66], and summarized with 'emmeans'[67] and 'sjPlot'[68], model residuals were checked for violations of assumptions using the 'DHARMa' package (https://github.com/Philip-Leftwich/Population-level-demonstration-of-multiplex-drive-Aedes-aegypti)[69,70]. **Mathematical modeling** Complete details of the model available in Supplementary information S2. All Matlab scripts used in the mathematical modeling are freely available via the Open Science Framework (osf.io/bp4yh).

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

## Acknowledgements
M.A.E.A., E.G., J.X.D.A., L.S., K.N., S.A.N.V., and L.A. were funded through a Defense Advanced Research Projects Agency (DARPA) award [N66001-17-2-4054] to Kevin Esvelt at MIT. MPE and PTL were supported by the Wellcome Trust [110117/Z/15/Z]. L.A. and T.H.S. were funded by the UK Biotechnology and Biological Sciences Research Council [BBS/E/I/00007033, BBS/E/I/00007038, and BBS/E/I/00007039 to The Pirbright Institute]. DKP's PhD studentship was funded by The Pirbright Institute. The views, opinions and/or findings expressed are those of the authors and should not be interpreted as representing the official views or policies of the U.S. Government. The funders had no role in study design, data collection and analysis, decision to publish, or preparation of the manuscript. For the purpose of Open Access, the author has applied a CC BY public copyright license to any Author Accepted Manuscript (AAM) version arising from this submission. We would like to thank Graham Freimanis at the Bioinformatics, Sequencing and Proteomics core facility for running the Illumina MiSeq, and advice and consultations with regards to the data. We would also like to thank Rebecka Ireland, Jessica Mavica, and Sophia Fochler for their assistance in the Insectary in the early stages of this project.

## Author contributions
M.A.E.A., T.H.S., P.T.L., K.E. and L.A. designed the research. M.A.E.A., E.G., D.K.P., J.X.D.A., L.S. and K.N. performed the research. M.A.E.A., T.H.S., P.T.L., S.A.N.V. contributed reagents. D.K.P., J.X.D.A., P.T.L., M.P.E. contributed analytic tools and analyzed the data. All authors contributed to writing and editing the paper.

## Competing interests
L.A. is an adviser to Synvect Inc and Biocentis Ltd, with equity and/or financial interest in those companies. The other authors declare that they have no competing interests.
