## [Peer Review File · Nature Communications]

REVIEWER COMMENTS

Reviewer #1 (Remarks to the Author):

In their manuscript, "A multiplexed, confinable CRISPR/Cas9 gene drive propagates in caged *Aedes aegypti* populations," Anderson, Gonzalez and colleagues describe an effective and novel strategy towards an effective super-Mendelian 'gene drive' in the yellow fever mosquito *Aedes aegypti*. This is a worthy and important goal that has eluded those working in *Ae. aegypti* as compared to colleagues working in *Anopheles gambiae*. Of note, increasing the rate of HDR as compared to NHEJ has been a difficult challenge for the yellow fever mosquito. The work presented here is of high quality and represents a tremendous effort at the bench and should be published - however, whether this particular set of reagents represents a promising 'gene drive' for the yellow fever mosquito is less clear.

The introduction to the manuscript nicely lays out the history of gene drive in organisms such as yeast and the malaria mosquito, while outlining problems that include the development of resistance alleles (especially the so-called r1 'silent' alleles) and the design principles behind a 'split drive' system. Notably, they find that the cross between a multiplexed kmo^ΔsgRNA cassette and bgcn-Cas9 resulted in a majority of cuts being repaired by HDR; this is a significant step up from past efforts. These data are important for the field as a whole and represent a step forward, however it's not clear yet whether gene drive has 'arrived' as a viable control strategy for *Aedes aegypti* mosquitoes. Thus, in order for this manuscript to be maximally useful for the field, some key details should be expanded upon (see major comments below).

Major comments:

- 1) It is not clear how the present manuscript arrived at testing bgcn as an ideal regulator of Cas9 activity. Was this part of a screen? In the past, these authors have characterized shu- and sds3-Cas9 lines that were effective (at least in some respects) - so why change now?
- 2) Similarly, the use of novel U6 promoters and 3 verified and 1 novel sgRNA introduces several degrees of freedom. Do the authors have a stance on which of the U6 promoters are likely to be most highly active in a general sense? Was U6.774-sRNA519 a 'dud' because of the U6 promoter or because of the sgRNA sequence itself?
- 3) The insect codon-optimized Cas9 appears to be novel to this work - more details on how it was designed and generated should be provided in the methods
- 4) The modelling results are presented before the sequencing results from F5 which indicate that some, but not all, animals have recognition sites intact for at least 2 sgRNAs. While the modelling is not this reviewer's area of expertise, the availability of intact sgRNAs seems like an important consideration when determining the long-term efficacy of the gene drive strategy, and this should be discussed
- 5) Upon reading this manuscript, it remains unclear to me how much of a 'home-run' this study represents. As the authors point out on line 365-367, these data indicate substantial improvement in inheritance bias as compared to previous studies - however, is it good enough for thinking about semi-

field or even field trials? The authors also acknowledge in lines 427-429 that 'further studies... could improve inheritance rates and decrease NHEJ...' Given this, I found missing from the discussion a thorough description of the authors' view on what hurdles remain to be leapt? How close are we to a gene drive for *Ae. aegypti* that is field-ready?

Minor comments:

6) Fig 1 - chromosomal coordinates should be added to give context for the kmo gene model

7) Fig S4 - in addition to naming the ladder in the legend, key band sizes could be added (also the sizes of the predicted products from the genotyping PCR)

8) Results - there is a thorough description of why the authors think that repair resulting from single cuts predominate the sequencing results (223-244) - they do acknowledge that larger-scale deletions or rearrangements are possible due to the fact that they would disrupt the PCR primer-binding sites used to characterize deletions, so this reviewer believes that the statement in lines 234-235 should be toned down accordingly.

Signed, Ben Matthews

Reviewer #2 (Remarks to the Author):

The authors describe a gene-drive system in *Aedes* using multiple gRNAs (multiplexing approach) to mitigate the formation of resistant alleles while improving gene drive propagation. The multiplexing approach has been used in *Drosophila* where no good spreading of the engineered alleles was observed. This work is novel per se as multiplexing approaches have not been developed in *Aedes* mosquitoes, as far as I know. I know of two works in *Anopheles* mosquitoes, and they were not as efficient as the traditional gene drive.

Please, see my comments below:

1. The authors state that "This avoids the outermost sgRNAs having a privileged position relative to the internal ones (without the requirement for resection) and addresses the critical question of whether this multiplexing design leads to a dramatic reduction in drive efficiency as has been demonstrated in *D. melanogaster* (41)"

To truly test this hypothesis, in my opinion, an additional gene drive system with extended homology arms and cut sites matching the engineered allele's homology arms during HDR is needed. Although their design employs shorter homology arms compared to the conventional approach, it's crucial to acknowledge that the existing data does not suffice for drawing conclusive differences. What we can assert is that multiplexing strategies employing shorter homology arms can achieve HDR at certain degree.

2. In reference to lines 84-87, it would greatly enhance clarity if you could state that, despite the theoretical viability of multiplexing approaches, its practical implementation in *Drosophila* experiments were not successful at spreading the engineered allele in caged populations.

3. For the sake of improved visual comprehension, it would be beneficial if you could incorporate homology arms as distinct features or boxes, mirroring the way UTR/Cyan and gRNAs are delineated in Figure 1. It will be also interesting to show the lack of homology between the engineered allele and the WT to better highlight the specifics of the design.

4. While the utilization of distinct U6 promoters to express the four distinct gRNAs potentially aids in minimizing homology within the drive allele to avoid weird events during HDR, it remains unclear if there is a specific rationale behind assigning each U6 promoter to a particular position (or gRNA). Unless I missed it, which is possible, it would greatly enhance the clarity of the design if the authors could explain their reasoning behind this specific configuration.

5. The current study encompasses two cage experiments (Fig.4), each yielding different outcomes. For greater robustness in the conclusions drawn, incorporating a third cage experiment would be ideal. Moreover, the retention of these cages for only six generations (as depicted in Fig. 4) raises certain concerns, given the standard practice of keeping them during more generations, especially when the presence of the gene drive seems declining at generation 6 in both (A1 and A2) cages. The inclusion of control cages, which assess the behavior of gRNAs (gene drive) and Cas9 in combination with wildtype individuals, is appreciated.

6. The success of multiplexing strategies hinges upon the temporal coordination of gRNAs. A deletion spanning all four gRNAs wasn't observed; instead, the predominant instances were single cuts or minor deletions stemming from distinct gRNAs. This suggests that the gRNAs might not function simultaneously or exhibit varying degrees of activity. Notably, line 340-342 shows the potential influence of gRNA site 519, possibly contributing to the reduced HDR efficiency. It would be valuable for the discussion section to delve into the interaction between gRNA cutting patterns and their impact on HDR levels.

7. Fig 3: A figure that illustrates the actual deletions would be more informative than solely depicting mutational rates. By supplementing this representation with percentages of each indel, the reader's comprehension would be significantly enhanced. Additionally, did the authors undertake a comparison between indels resulting from single crosses and those emerging from cage trials? Such a juxtaposition could reveal potential disparities in their dynamics.

8. Has the sequencing of Cyan+ individuals from cage experiments been undertaken by the authors? Exploring the possibility of DNA rearrangements that could introduce Cyan+ mosquitoes into the population without the complete gene drive element is an interesting point to keep in mind. This can be readily assessed through PCR and gel electrophoresis. While the assumption is often that the fluorescent marker signifies the entire gene drive, multiplexing approaches introduce multiple cuts and could produce unexpected outcomes.

9. The title "A multiplexed, confinable CRISPR/Cas9 gene drive propagates in caged *Aedes aegypti* populations" might benefit from toning down as it does not fully propagate. This study is limited to two cages, with the gene drives exhibiting approximately 75% and 60% transmission rates by generation 6.

Overall, I think this work is novel and interesting from a technical point of view as multiplexing approaches have not been deeply explored in mosquitoes. Nonetheless, the limitation of maintaining cages for only six generations ($n=2$), along with the observed fitness costs, warrants consideration for a more extensive analysis.

The authors would like to thank the reviewers for their time and insightful comments. We have addressed all comments in the point by point response below.

REVIEWER COMMENTS

Reviewer #1 (Remarks to the Author):

In their manuscript, “A multiplexed, confinable CRISPR/Cas9 gene drive propagates in caged *Aedes aegypti* populations,” Anderson, Gonzalez and colleagues describe an effective and novel strategy towards an effective super-Mendelian ‘gene drive’ in the yellow fever mosquito *Aedes aegypti*. This is a worthy and important goal that has eluded those working in *Ae. aegypti* as compared to colleagues working in *Anopheles gambiae*. Of note, increasing the rate of HDR as compared to NHEJ has been a difficult challenge for the yellow fever mosquito. The work presented here is of high quality and represents a tremendous effort at the bench and should be published - however, whether this particular set of reagents represents a promising ‘gene drive’ for the yellow fever mosquito is less clear.

The introduction to the manuscript nicely lays out the history of gene drive in organisms such as yeast and the malaria mosquito, while outlining problems that include the development of resistance alleles (especially the so-called r1 ‘silent’ alleles) and the design principles behind a ‘split drive’ system. Notably, they find that the cross between a multiplexed kmo^ΔsgRNA cassette and bgcn-Cas9 resulted in a majority of cuts being repaired by HDR; this is a significant step up from past efforts. These data are important for the field as a whole and represent a step forward, however it’s not clear yet whether gene drive has ‘arrived’ as a viable control strategy for *Aedes aegypti* mosquitoes. Thus, in order for this manuscript to be maximally useful for the field, some key details should be expanded upon (see major comments below).

Major comments:

1) It is not clear how the present manuscript arrived at testing bgcn as an ideal regulator of Cas9 activity. Was this part of a screen? In the past, these authors have characterized shu- and sds3-Cas9 lines that were effective (at least in some respects) - so why change now?

At the time the cage trial was initiated the alternative promoters had not yet been characterised. We agree that the shu and sds3-Cas9 lines may be more effective – or indeed other, yet-to-be-identified promoters. Unfortunately funding and the timescale of the project did not allow us to undertake additional cage trials with these other promoters. A reference to the shu/SDS3 article has been included in the discussion.

2) Similarly, the use of novel U6 promoters and 3 verified and 1 novel sgRNA introduces several degrees of freedom. Do the authors have a stance on which of the U6 promoters are likely to be most highly active in a general sense? Was U6.774-sRNA519 a ‘dud’ because of the U6 promoter or because of the sgRNA sequence itself?

*More information regarding the relative strength of the promoters and activity of the guides (from previously published works) has been included in the results section where the design of the kmo^{sgRNAs} element is described; this has also been addressed in the discussion. It was surprising that 519 was a bit of a dud as this sgRNA was previously tested by others and shown to be the most active as determined by embryo injection followed by HRMA. It was then selected for use in generating some of the first CRISPR/Cas9 gene edited mosquitoes (Ref 39 in the manuscript, Basu et al PNAS 2015). Similarly the pol III promoter expressing this sgRNA (U6.774) was also assessed by others in *Aedes aegypti* transgenics and it was the second most active – and not much below the most active (Ref 38 in the manuscript, Li et al eLife 2020). We suspect there is some position effect on pol III promoter expression but this has not been thoroughly investigated by us or others that we are aware of.*

3) The insect codon-optimized Cas9 appears to be novel to this work - more details on how it was designed and generated should be provided in the methods

This section of the methods has been expanded to contain more details for the Cas9.

4) The modelling results are presented before the sequencing results from F5 which indicate that some, but not all, animals have recognition sites intact for at least 2 sgRNAs. While the modelling is not this reviewer's area of expertise, the availability of intact sgRNAs seems like an important consideration when determining the long-term efficacy of the gene drive strategy, and this should be discussed

We have interpreted this comment as referring to the importance of intact target sites on the long-term performance of the gene drive - particularly focussing on the inclusion of this within the mathematical modelling shown in this study.

Within this study we have only considered the first six generations since the gene drive carrier frequency is already in decline by this point. Since the models shown largely agree with the experimental results, this would suggest that the availability of intact target sites can be neglected over this relatively short timescale.

However, since we agree with the Reviewer that the availability of intact target site would be extremely important in determining the efficacy of gene drive systems over longer timescales, we have added the below comment to the end of the mathematical modelling section of the manuscript.

“Note that changes in the availability of intact target sgRNA target sites have been neglected within the mathematical model due to the relatively short timescale considered - which is supported by the broad agreement between the models and experimental data. However, we would expect this to be of great importance when modeling the efficacy of gene drive systems that persist over longer timescales.”

5) Upon reading this manuscript, it remains unclear to me how much of a 'home-run' this study represents. As the authors point out on line 365-367, these data indicate substantial improvement in inheritance bias as compared to previous studies - however, is it good enough for thinking about semi-field or even field trials? The authors also acknowledge in lines 427-429 that 'further studies... could improve inheritance rates and decrease NHEJ...' Given this, I found missing from the discussion a thorough description of the authors' view on what hurdles remain to be leapt? How close are we to a gene drive for *Ae. aegypti* that is field-ready?

We agree with the sentiment of the reviewer – this is real progress but we still aren't there yet in terms of a field usable gene drive in this species. The discussion has been expanded to include what we think remains to be improved on before a gene drive is feasible in this species.

Minor comments:

6) Fig 1 - chromosomal coordinates should be added to give context for the kmo gene model

Coordinates have been added.

7) Fig S4 - in addition to naming the ladder in the legend, key band sizes could be added (also the sizes of the predicted products from the genotyping PCR)

This information has been added to the Figure and legend.

8) Results - there is a thorough description of why the authors think that repair resulting from single cuts predominate the sequencing results (223-244) - they do acknowledge that larger-scale deletions or rearrangements are possible due to the fact that they would disrupt the PCR primer-binding sites used to characterize deletions, so this reviewer believes that the statement in lines 234-235 should be toned down accordingly.

This statement has been revised.

Signed, Ben Matthews

Reviewer #2 (Remarks to the Author):

The authors describe a gene-drive system in *Aedes* using multiple gRNAs (multiplexing approach) to mitigate the formation of resistant alleles while improving gene drive propagation. The multiplexing approach has been used in *Drosophila* where no good spreading of the engineered alleles was observed. This work is novel per se as multiplexing approaches have not been developed in *Aedes* mosquitoes, as far as I know. I know of two works in *Anopheles* mosquitoes, and they were not as efficient as the traditional gene drive.

Please, see my comments below:

1. The authors state that “This avoids the outermost sgRNAs having a privileged position relative to the internal ones (without the requirement for resection) and addresses the critical question of whether this multiplexing design leads to a dramatic reduction in drive efficiency as has been demonstrated in *D. melanogaster* (41)”

To truly test this hypothesis, in my opinion, an additional gene drive system with extended homology arms and cut sites matching the engineered allele's homology arms during HDR is needed. Although their design employs shorter homology arms compared to the conventional approach, it's crucial to acknowledge that the existing data does not suffice for drawing conclusive differences.

What we can assert is that multiplexing strategies employing shorter homology arms can achieve HDR at certain degree.

This section has been rephrased.

2. In reference to lines 84-87, it would greatly enhance clarity if you could state that, despite the theoretical viability of multiplexing approaches, its practical implementation in *Drosophila* experiments were not successful at spreading the engineered allele in caged populations.

This section has been expanded to provide more details.

3. For the sake of improved visual comprehension, it would be beneficial if you could incorporate homology arms as distinct features or boxes, mirroring the way UTR/Cyan and gRNAs are delineated in Figure 1. It will be also interesting to show the lack of homology between the engineered allele and the WT to better highlight the specifics of the design.

This figure has been revised for clarity.

4. While the utilization of distinct U6 promoters to express the four distinct gRNAs potentially aids in minimizing homology within the drive allele to avoid weird events during HDR, it remains unclear if there is a specific rationale behind assigning each U6 promoter to a particular position (or gRNA). Unless I missed it, which is possible, it would greatly enhance the clarity of the design if the authors could explain their reasoning behind this specific configuration.

Additional information regarding the pol III promoters and sgRNAs has been added to the results section.

5. The current study encompasses two cage experiments (Fig.4), each yielding different outcomes. For greater robustness in the conclusions drawn, incorporating a third cage experiment would be ideal. Moreover, the retention of these cages for only six generations (as depicted in Fig. 4) raises certain concerns, given the standard practice of keeping them during more generations, especially when the presence of the gene drive seems declining at generation 6 in both (A1 and A2) cages. The inclusion of control cages, which assess the behavior of gRNAs (gene drive) and Cas9 in combination with wildtype individuals, is appreciated.

*In regards to the number of replicate cages in the trial, we selected to perform this in duplicate for several reasons. Firstly, many publications with similar cage trials performed with *Anopheles* gene drives in very high-impact journals were performed in duplicate:*

*Hammond, A., Galizi, R., Kyrou, K. et al. A CRISPR-Cas9 gene drive system targeting female reproduction in the malaria mosquito vector *Anopheles gambiae*. *Nat Biotechnol* 34, 78–83 (2016). <https://doi.org/10.1038/nbt.3439>*

*Hammond AM, Kyrou K, Bruttini M, North A, Galizi R, Karlsson X, et al. (2017) The creation and selection of mutations resistant to a gene drive over multiple generations in the malaria mosquito. *PLoS Genet* 13(10): e1007039. <https://doi.org/10.1371/journal.pgen.1007039>*

*Kyrou, K., Hammond, A., Galizi, R. et al. A CRISPR-Cas9 gene drive targeting doublesex causes complete population suppression in caged *Anopheles gambiae* mosquitoes. *Nat Biotechnol* 36, 1062–1066 (2018). <https://doi.org/10.1038/nbt.4245>*

*Simoni, A., Hammond, A.M., Beaghton, A.K. et al. A male-biased sex-distorter gene drive for the human malaria vector *Anopheles gambiae*. *Nat Biotechnol* 38, 1054–1060 (2020). <https://doi.org/10.1038/s41587-020-0508-1>*

Hammond, A., Pollegioni, P., Persampieri, T. et al. Gene-drive suppression of mosquito populations in large cages as a bridge between lab and field. *Nat Commun* 12, 4589 (2021).

<https://doi.org/10.1038/s41467-021-24790-6>

Taxiarchi, C., Beaghton, A., Don, N.I. et al. A genetically encoded anti-CRISPR protein constrains gene drive spread and prevents population suppression. *Nat Commun* 12, 3977 (2021).

<https://doi.org/10.1038/s41467-021-24214-5>

Hammond A, Karlsson X, Morianou I, Kyrou K, Beaghton A, Gribble M, et al. (2021) Regulating the expression of gene drives is key to increasing their invasive potential and the mitigation of resistance. *PLoS Genet* 17(1): e1009321. <https://doi.org/10.1371/journal.pgen.1009321>

These other recent publications use triplicate cages in their cage trials:

Pham TB, Phong CH, Bennett JB, Hwang K, Jasinskiene N, Parker K, et al. (2019) Experimental population modification of the malaria vector mosquito, *Anopheles stephensi*. *PLoS Genet* 15(12): e1008440. <https://doi.org/10.1371/journal.pgen.1008440>

Adolfi, A., Gantz, V.M., Jasinskiene, N. et al. Efficient population modification gene-drive rescue system in the malaria mosquito *Anopheles stephensi*. *Nat Commun* 11, 5553 (2020).

<https://doi.org/10.1038/s41467-020-19426-0>

Carballar-Lejarazú, R., Pham, T. B., Bottino-Rojas, V., Adolfi, A., James, A. A. Small-Cage Laboratory Trials of Genetically-Engineered Anopheline Mosquitoes. *J. Vis. Exp.* (171), e62588, [doi:10.3791/62588](https://doi.org/10.3791/62588) (2021).

We had to balance the high time commitment required to perform these trials, with the total number of cages for the study, to ensure we also could run the appropriate control cages. As one would expect, each cage run produced somewhat different outcomes, as indeed do independent runs of the stochastic model. We performed two replicates of the full split drive (Cages A1 and A2) - in one of these replicates (A2) endpoints for both transgenes overlap with the mean endpoint generated by the model to such an extent that they are indistinguishable. For the other replicate cage (A1) the cas9 transgene endpoint is exactly as the model endpoint, while the A element endpoint falls within the expected variance of the model for the majority of the experiment (markedly deviating from this only in the final of 6 generations).

6. The success of multiplexing strategies hinges upon the temporal coordination of gRNAs. A deletion spanning all four gRNAs wasn't observed; instead, the predominant instances were single cuts or minor deletions stemming from distinct gRNAs. This suggests that the gRNAs might not function simultaneously or exhibit varying degrees of activity. Notably, line 340-342 shows the potential influence of gRNA site 519, possibly contributing to the reduced HDR efficiency. It would be valuable for the discussion section to delve into the interaction between gRNA cutting patterns and their impact on HDR levels.

The discussion has been expanded to include these points.

7. Fig 3: A figure that illustrates the actual deletions would be more informative than solely depicting mutational rates. By supplementing this representation with percentages of each indel, the reader's comprehension would be significantly enhanced. Additionally, did the authors undertake a comparison between indels resulting from single crosses and those emerging from cage trials? Such a juxtaposition could reveal potential disparities in their dynamics.

For this experiment we have crossed transheterozygotes to our kmo KO line. This line however, is a pool of several different mutations. When analysing this data it is impossible to sort out which mutation was generated in the F2 individual and which mutation was inherited from the kmo KO parent. We also observed mosaicism from deposited Cas9 in the progeny of female

transheterozygotes, generating a range of mutations even within a single individual. Thus from this dataset we determined we could merely gain insight into the mutational rates for each sgRNA, rather than the sequence of the mutations generated.

8. Has the sequencing of Cyan+ individuals from cage experiments been undertaken by the authors? Exploring the possibility of DNA rearrangements that could introduce Cyan+ mosquitoes into the population without the complete gene drive element is an interesting point to keep in mind. This can be readily assessed through PCR and gel electrophoresis. While the assumption is often that the fluorescent marker signifies the entire gene drive, multiplexing approaches introduce multiple cuts and could produce unexpected outcomes.

While it is true that PCR and gel electrophoresis could be performed on Cyan+ individuals from the cage trial, such analysis was not undertaken in this study for several reasons. Firstly, the majority of Cyan+ individuals would have inherited the transgene through standard inheritance, rather than through homing events and thus a large number of individuals would need to be screened to identify the homed alleles. Secondly, such large-scale DNA rearrangements such as those the Reviewer suggests would also likely disrupt primer binding sites and would correspondingly be difficult to systematically identify and analyse in a PCR-based assay.

9. The title "A multiplexed, confinable CRISPR/Cas9 gene drive propagates in caged *Aedes aegypti* populations" might benefit from toning down as it does not fully propagate. This study is limited to two cages, with the gene drives exhibiting approximately 75% and 60% transmission rates by generation 6.

We selected the term "propagate" rather than others such as "drives" or "proliferates" as we felt it was toned down from those terms as it indicates spread rather than complete fixation. However, we have further revised this to "can propagate".

Overall, I think this work is novel and interesting from a technical point of view as multiplexing approaches have not been deeply explored in mosquitoes. Nonetheless, the limitation of maintaining cages for only six generations ($n=2$), along with the observed fitness costs, warrants consideration for a more extensive analysis.

REVIEWERS' COMMENTS

Reviewer #1 (Remarks to the Author):

After reading the updated manuscript and the response to reviewers, I am satisfied that this work represents a significant step forward in the generation of gene drives in *Aedes aegypti* mosquitoes. The authors have considered all points and present a thoughtful and thorough discussion of this work in the broader context of generating gene drives in this tricky species.

Reviewer #2 (Remarks to the Author):

The authors have addressed all the comments and expanded on some interesting points throughout the discussion. As I mentioned, I find this work interesting from a technical standpoint, and I believe it will contribute to future gene drive designs that aim to implement multiplexing approaches in mosquitoes and beyond.